# Reproductive Characteristics of the Flat Oyster *Ostrea denselamellosa* (Bivalvia, Ostreidae) Found on the Southern Coast of South Korea

**Jeonghoon Han, Han-Jun Kim, Sung-Yong Oh and Young-Ung Choi ***

Marine Bio-Resources Research Unit, Korea Institute of Ocean Science & Technology (KIOST), Busan 49111, Korea
* Correspondence: yuchoi@kiost.ac.kr

**Abstract:** In this study, we investigated the reproductive pattern of the commercially and ecologically important species, *Ostrea denelamellosa*, to inform stock management strategies in South Korea. Prior to the reproduction experiment, the complete mitochondrial (mt) genome of the flat oyster, *Ostrea denselamellosa*, was analyzed using next-generation sequencing technology. Then, to determine the reproductive pattern of *Ostrea denselamellosa*, we investigated monthly changes in the gametogenesis, reproductive cycle, and sex ratio from January to October 2021 in females. The total length of the mt genome sequence of *O. denselamellosa* was 16,225 bp and contained 37 genes (13 protein-coding genes, 22 tRNA genes, and 2 rRNA genes). Molecular phylogenetic comparison with 20 known species of Pteriomorphia showed that *O. denselamellosa* belongs to the family Ostreidae. In addition, *O. denselamellosa* clustered together with the *O. denselamellosa* Chinese strain, with a bootstrap value of 100%. Histological analysis indicated a discrepancy in gamete development of *O. denselamellosa* with synchronous maturation of oocytes and asynchronous development of spermatozoa in gonads. The spawning activity occurred between May and September with a temperature range gap of 6.5 °C. The spawning activity occurred from May when the temperature reached 16.7 °C until September when the temperature dropped below 23.2 °C. Furthermore, sex ratio bias was observed. This is the first study to report the complete mt genome sequence and examine the reproductive pattern in native *O. denselamellosa* in South Korea. Overall, these findings will help enhance the knowledge for the management and sustainable fishery of endangered oyster species including *O. denselamellosa* in the South Sea of Korea.

**Keywords:** bivalve; marine resources; phylogeny; environmental factors; gametogenesis; gonad development; sex differentiation

## 1. Introduction

Oysters represent economically important species as marine resources in fishery and aquaculture and are ecologically important because they provide critical ecosystem services as habitat engineers, calcifiers, filter-feeders, and reef-builders [1]. The annual production of global oyster aquaculture is approximately 5.9 million metric tons [2]. However, oyster populations have declined owing to pollution, overfishing, habitat loss, and diseases, resulting in adverse effects such as reduced water quality and loss of biodiversity [3–6].

The flat oyster *Ostrea denselamellosa* is commonly found in subtidal mudflats off the southwestern and southern coastlines of Korea [7,8] as well as in southern Japan and China [9,10]. Presently, *O. denselamellosa* is considered an important bivalve resource in coastal shellfish fisheries [10]. However, natural populations of *O. denselamellosa* have declined dramatically due to anthropogenic effects, including over-exploitation and environmental pollution [11], thus the population density of *O. denselamellosa* is known to be low [10]. However, studies on *O. denselamellosa* have mainly focused on cultural methods and biological characteristics [11,12]. Therefore, understanding genetic diversity and population density are essential for the management and sustainable use of oyster resources.

In this regard, mitochondrial genomes (mt genomes) have been extensively studied as molecular markers for species identification, population genetics, conservation biology, and diverse evolutionary studies [13–15]. In particular, sequencing technology for complete animal mt genomes has been used for phylogenetic reconstruction compared to using partial sequences. Therefore, complete mt genomes can be applied to genetic molecular marker-based species identification, population genetics, conservation biology, and diverse evolutionary studies. A previous study reported the complete mt genome of *O. denselamellosa* collected from Jiaonan, Shandong Province, China [16]. However, there is no information on the complete mt genome of *O. denselamellosa* native oysters in South Korea.

Generally, the reproductive patterns of oysters in the *Ostrea* genus involve spermcasting, which is the internal cross-fertilization of retained eggs via the release, dispersal, and uptake of free spermatozoa [17,18]. In addition, *O. denselamellosa* is viviparous and deposits its larvae directly into the water column after a brooding period with fertilized eggs inside their body cavity [11,19]. In this regard, for achieving the sustainable management and use of oyster resources, many environmental factors must be considered. Various reproductive characteristics and strategies of marine bivalves, including oysters, are controlled by environmental factors such as water temperature and food availability [20–23]. In particular, water temperature is the main factor that controls various physiological processes, including the reproduction cycle in bivalves [21,24–26]. However, the characteristics of the reproductive biology of the native oyster in South Korea have not been elucidated. Therefore, as the initial step, we determined the complete mt genome sequence and phylogenetic relationships of native *O. denselamellosa* collected from South Korea. In addition, we investigated the gametogenesis, reproductive cycle, and sex ratio of *O. denselamellosa*.

## 2. Materials and Methods

### 2.1. Sampling Collection

The *O. denselamellosa* was purchased from a local fish market in Buan, South Korea, on 20 December 2020 and maintained in culture cage nets at a depth of 2 m from the surface at the cage farm at the Tongyoung marine living resources station, Korea Institute of Ocean Science and Technology (KIOST), Tongyeong, Gyeongsangnam-do, South Sea of Korea (Figure 1). Live *O. denselamellosa* were collected every month (20 oysters/month) from January 2021 to October 2021. At the time of sampling, the surface water temperature and salinity ranged between 11.4 and 26.1 °C and 29.0 and 33.4 practical salinity units, respectively (Figure 2). All animal experimental protocols were performed according to the Guidelines of the Institutional Animal Care and Experimental Committee and were approved by the KIOST.

### 2.2. DNA Extraction and DNA Sequencing

Genomic DNA was extracted from the muscle tissue of oysters using the DNeasy Blood and Tissue Kit (Qiagen, Valencia, CA, USA) following the manufacturer's protocol. The quantity and quality of isolated DNA were analyzed and measured at 230, 260, and 280 nm using a NanoDrop One spectrophotometer (Thermo Fisher Scientific, Madison, WI, USA). Whole-genome sequencing was performed using an Illumina NovaSeq 6000 platform (Illumina, San Diego, CA, USA) at the National Instrumentation Center for Environmental Management, Seoul, South Korea. The complete mt genome of *O. denselamellosa* was assembled and annotated using MitoZ [27].

### 2.3. Sequence Alignment and Phylogenetic Analysis

The complete mt genomes of 20 Pteriomorphia species were downloaded from the GenBank database and used for constructing a phylogenetic tree. The blacklip abalone *Haliotis rubra* (Gastropoda) mt genome was chosen as the outgroup (Table 1). The amino acid sequences of 12 protein-coding genes (PCGs) for each mt genome were aligned using the ClustalW algorithm in MEGA software (ver. 10.0.1; Center for Evolutionary Medicine and Informatics, Tempe, AZ, USA). To establish the best-fit substitution model for phylogenetic

analysis, the model with the lowest Bayesian Information Criterion and Akaike Information Criterion scores was estimated using a maximum-likelihood (ML) analysis. According to the results of the model test, the ML phylogenetic analyses were performed using the LG + G + I model in the MEGA software. The support for nodes was calculated using 1000 bootstrap replicates.

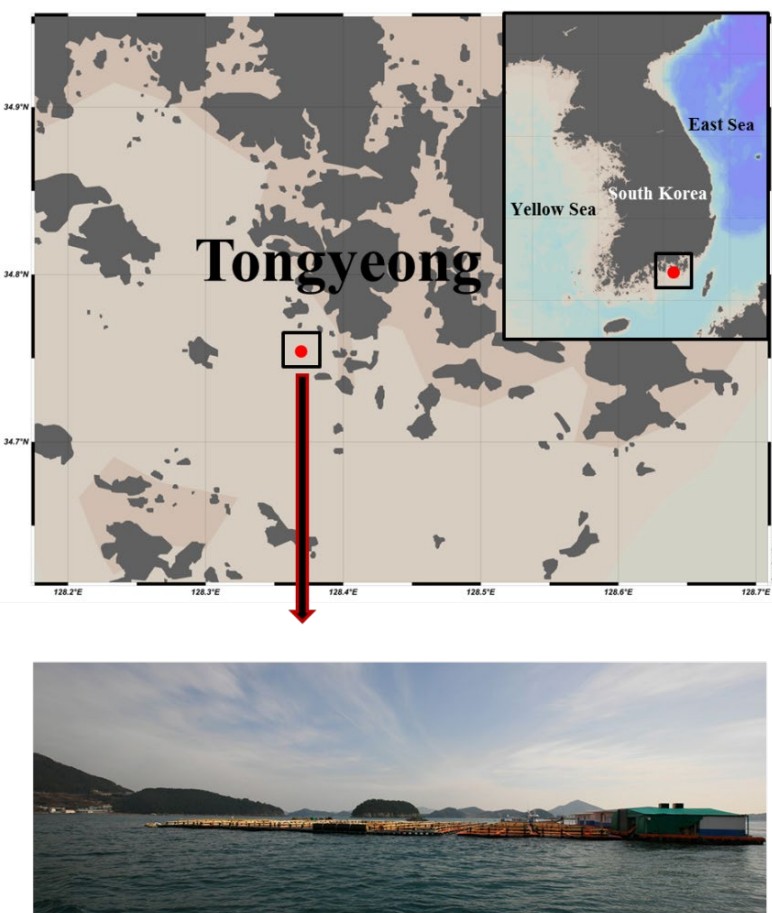

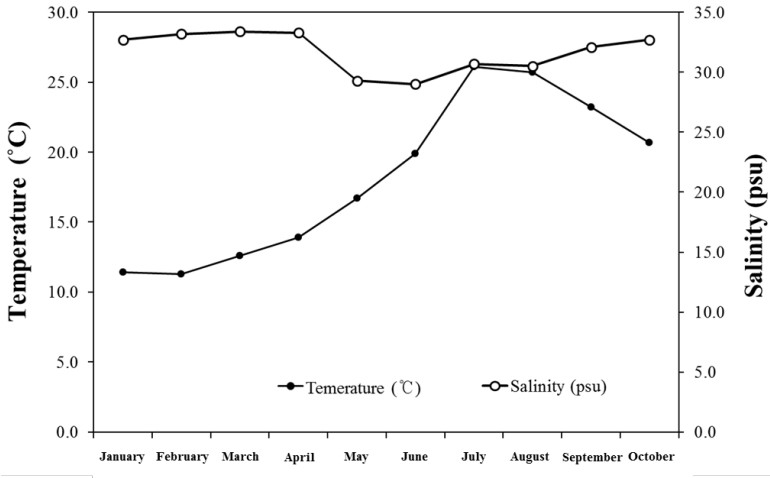

**Figure 1.** Map showing Tongyoung marine living resources station, Tongyeong, Gyeongsangnam-do, South Korea.

**Figure 2.** Monthly changes in the surface water temperature and salinity at the sampling site from January 2021 to October 2021. psu, practical salinity unit.

**Table 1.** List of complete mt genomes used in this study.

| | Tax On | Classification | Size (bp) | Accession No. |
|---|---|---|---|---|
| | *Mytilus edulis* | Mytiloida; Mytiloidea; Mytilidae | 16,740 | AY484747 |
| | *Mytilus galloprovincialis* | Mytiloida; Mytiloidea; Mytilidae | 16,744 | AY497292 |
| | *Mytilus trossulus* | Mytiloida; Mytiloidea; Mytilidae | 18,652 | AY823625 |
| | *Musculista senhousia* | Mytiloida; Mytiloidea; Mytilidae | 20,612 | GU001954 |
| | *Crassostrea angulata* | Ostreoida; Ostreoidea; Ostreidae | 18,225 | EU672832 |
| | *Crassostrea ariakensis* | Ostreoida; Ostreoidea; Ostreidae | 18,414 | EU672835 |
| | *Crassostrea gigas* | Ostreoida; Ostreoidea; Ostreidae | 18,225 | EU672831 |
| | *Crassostrea hongkongensis* | Ostreoida; Ostreoidea; Ostreidae | 18,622 | EU672834 |
| | *Crassostrea iredalei* | Ostreoida; Ostreoidea; Ostreidae | 22,446 | FJ841967 |
| **Mollusca** | *Crassostrea sikamea* | Ostreoida; Ostreoidea; Ostreidae | 18,243 | EU672833 |
| **Bivalvia** | *Saccostrea mordax* | Ostreoida; Ostreoidea; Ostreidae | 16,532 | FJ841968 |
| **Pteriomorphia** | *Saccostrea glomerata* | Ostreoida; Ostreoidea; Ostreidae | 16,281 | KU310918 |
| | *Ostrea denselamellosa* | Ostreoida; Ostreoidea; Ostreidae | 16,277 | HM015199 |
| | *Ostrea edulis* | Ostreoida; Ostreoidea; Ostreidae | 16,320 | JF274008 |
| | *Ostrea lurida* | Ostreoida; Ostreoidea; Ostreidae | 16,344 | KC768038 |
| | *Argopecten irradians* | Pectinoida; Pectinoidae; Pectinidae | 16,221 | EU023915 |
| | *Chlamys farreri* | Pectinoida; Pectinoidae; Pectinidae | 21,695 | EU715252 |
| | *Mizuhopecten yessoensis* | Pectinoida; Pectinoidae; Pectinidae | 20,414 | AB271769 |
| | *Placopecten magellanicus* | Pectinoida; Pectinoidae; Pectinidae | 32,115 | DQ088274 |
| | *Mimachlamys nobilis* | Pectinoida; Pectinoidae; Pectinidae | 17,963 | FJ415225 |
| **Gastropoda Vetigastropoda** | *Haliotis rubra* | Haliotoidea; Haliotidae | 16,907 | NC_005940 |

*2.4. Histological Analysis*

A total of 197 oysters were collected including 180 individuals from January 2021 to September 2021 and 17 individuals in October 2021. The monthly shell length (SL) and total weight (TW) were recorded during the experimental period. After oysters were anesthetized using 2 mL L$^{-1}$ propylene phenoxetol, the tissues were dehydrated in increasing ethanol concentrations, clarified in xylene, and embedded in paraffin. Sections (5 μm-thick) were stained with hematoxylin–eosin for observation under a light microscope (DM 100; Leica, Wetzlar, Germany) [28]; the images were captured using a digital camera (DFC 290; Leica). The gonads were classified into five stages according to a previous study [29], with slight modifications including (1) undifferentiated or resting gonad, (2) early gametogenesis, (3) advanced gametogenesis, (4) mature gonad, and (5) spawned gonad. Classification of sex categories (male, female, hermaphrodite, or undifferentiated) was recorded from January 2021 to October 2021.

**3. Results and Discussion**

In this study, we sequenced the complete mt genome of *O. denselamellosa* and analyzed its phylogenetic position (Table 2 and Figure 3). The length of the complete mt genome of *O. denselamellosa* was 16,275 bp (GenBank number: ON964460). This size is similar to that of *O. denselamellosa* (16,277 bp) collected from Jiaonan, Shandong Province, China [16] but shorter than those of *Ostrea edulis* (European flat oyster; 16,320 bp) and *Ostrea lurida* (Olympia oyster; 16,344 bp) [30,31]. The complete mt genome of *O. denselamellosa* contained 13 PCGs, 2 rRNA genes, and 23 tRNAs (Table 2 and Figure 3A), whereas 12 PCGs (without atp8), 2 rRNA genes, and 23 tRNA genes were previously identified in the Chinese strain [16]. Possible strain-specific differences may be due to compositional differences. For example, the A + T and G + C compositions of 13 PCGs in the mt genome of *O. denselamellosa* were 59.89% and 40.11%, respectively, whereas these compositions in all sequences were 60.59% and 39.41%, respectively. In particular, the ratio of A + T nucleotides in the mt genome of *O. denselamellosa* is similar to that of the *O. denselamellosa* Chinese strain (61%), whereas the ratio of A + T nucleotides is lower than those of the congeneric species *O. edulis* (64.9%) and *O. lurida* (65%) [16,30,31]. In *O. denselamellosa*, the ten PCGs initiate with the

start codon ATG/ATA, whereas atp6 and nd4l have the start codon GTG. Most of the PCGs (11 of 13 genes) terminate with TAA/TAG, whereas atp6 terminates with CGT. In contrast to *O. denselamellosa*, ten PCGs initiate with the start codon ATG/ATA, whereas atp6 and nd4l have the start codon GTG in the O. denselamellosa Chinese strain. Moreover, most PCGs (11 of 12 genes) terminate with TAA/TAG, whereas cox3 terminates with T– –. Therefore, comparative mt genome analysis of *O. denselamellosa* revealed the species and region-specific differences in the mt genomes of *Ostrea* species including the two strains of *O. denselamellosa*. Furthermore, minor differences in sequence identity may contribute to their adaptability to different environmental conditions; however, the adaptability-related potential requires further analysis. Furthermore, the region-specific speciations of *O. dense-lamellosa* could provide information about how reproductive strategies differ depending on their adaptation to the environment.

**Table 2.** Summary of *Ostrea denselamellosa* mitogenome.

| Gene | Location | Size (bp) | Start Codon | Stop Codon | Intergenic Region * |
|---|---|---|---|---|---|
| ATP6 | 1–550 | 550 | GTG | CGT | 0 |
| trnY | 551–613 | 63 | - | - | 2 |
| trnC | 616–678 | 62 | - | - | 36 |
| ND2 | 716–1714 | 999 | ATG | TAA | 37 |
| trnP | 2336–2399 | 64 | - | - | 621 |
| trnL | 2401–2467 | 67 | - | - | 1 |
| trnS1 | 2468–2537 | 70 | - | - | 0 |
| trnM1 | 2538–2600 | 63 | - | - | 0 |
| ATP8 | 2625–2726 | 102 | ATA | TAA | 24 |
| trnS2 | 2745–2814 | 70 | | | 18 |
| trnM2 | 2821–2883 | 63 | | | 6 |
| COX2 | 2889–3585 | 696 | ATG | TAG | 1 |
| CYTB | 3587–4747 | 1161 | ATA | TAA | 1 |
| trnE | 4745–4812 | 67 | - | - | 8 |
| trnT | 4821–4885 | 65 | - | - | 8 |
| trnI | 4894–4960 | 67 | - | - | 18 |
| COX3 | 4941–5840 | 900 | ATG | TAA | 130 |
| trnG | 5971–6037 | 67 | - | - | 8 |
| COX1 | 6046–7641 | 1596 | ATG | TAA | 688 |
| trnD | 8330–8397 | 68 | - | - | 55 |
| trnW | 8453–8515 | 63 | - | - | 60 |
| ND4L | 8576–8857 | 282 | GTG | TAA | 1 |
| ND1 | 8859–9791 | 933 | ATG | TAA | 78 |
| trnA | 9870–9934 | 65 | - | - | 12 |
| trnF | 9947–10,013 | 67 | - | - | 2 |
| trnL | 10,016–10,081 | 66 | - | - | 2 |
| trnK | 10,084–10,148 | 65 | - | - | −1 |
| ND3 | 10,148–10,501 | 354 | ATG | TAG | 0 |
| trnQ | 10,502–10,567 | 66 | - | - | −1 |
| ND6 | 10,579–11,046 | 468 | ATG | TAA | −1 |
| ND5 | 11,046–12,716 | 1671 | ATG | TAA | 37 |
| l-rRNA | 12,754–14,221 | 1468 | - | - | −660 |
| s-rRNA | 13,561–14,488 | 928 | - | - | 0 |
| ND4 | 14,495–15,844 | 1350 | ATG | TAA | 24 |
| trnH | 15,845–15,907 | 63 | - | - | 24 |
| trnV | 15,932–15,999 | 68 | - | - | 24 |
| trnR | 16,008–16,074 | 67 | - | - | 4 |
| trnN | 16,079–16,147 | 69 | - | - | - |

* Negative numbers indicate overlapping nucleotides between adjacent genes.

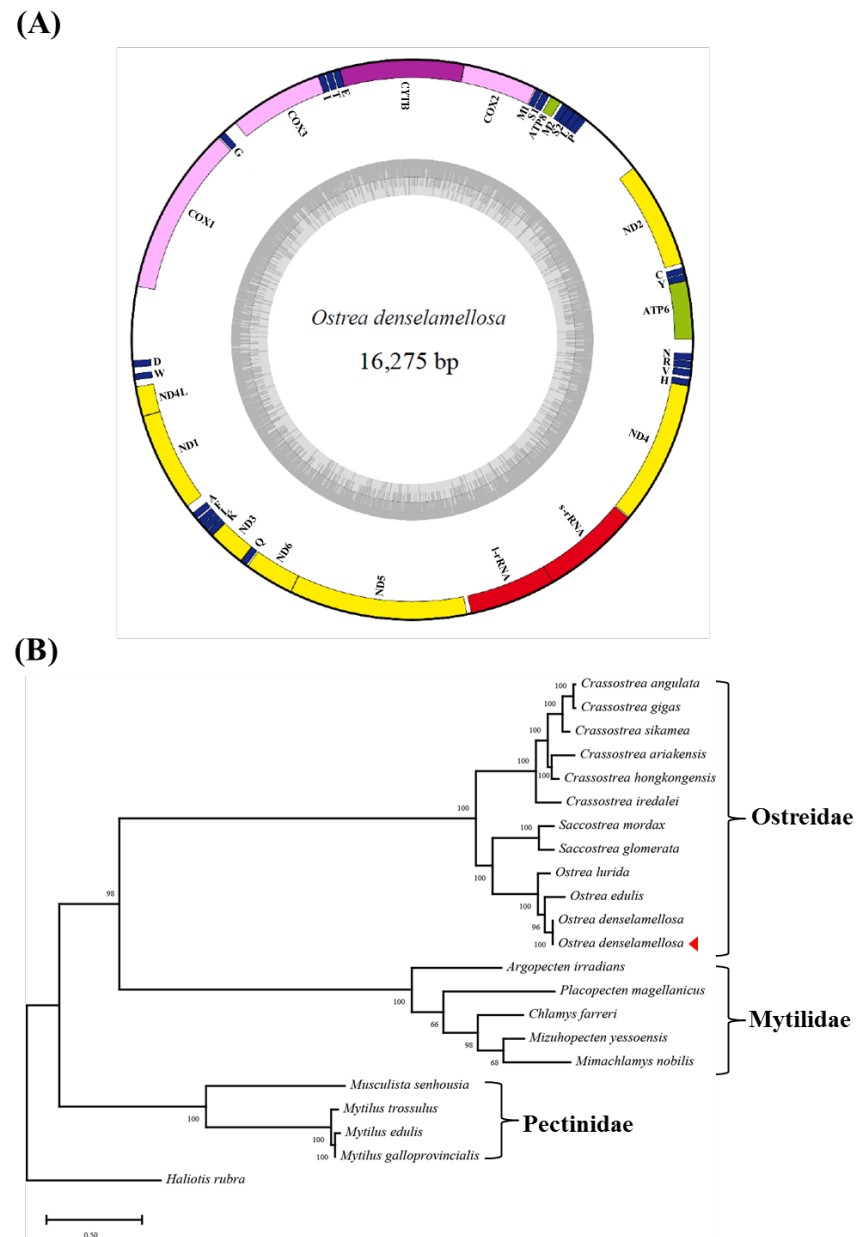

**Figure 3.** (**A**) The mitochondrial genome map of *Ostrea denselamellosa* and (**B**) Maximum-likelihood phylogeny of 21 published complete mitochondrial genomes based on 12 concatenated nucleotide sequences of protein-coding genes. The red triangle indicates the *O. denselamellosa* strain analyzed in this study.

In this study, the overall topology was consistent with previous phylogenetic results [30]. The molecular phylogenetic tree based on 13 PCG sequences showed that the *O. denselamellosa* Korean strain clustered together with three *Ostrea* species (*O. denselamellosa* Chinese strain, *O. edulis*, and *O. lurida*) (Figure 3B). In particular, *O. denselamellosa* clustered together with the *O. denselamellosa* Chinese strain with a bootstrap value of 100% (Figure 3B), indicating that *O. denselamellosa* is a congeneric species to the *O. denselamellosa* Chinese strain. Taken together, the newly completed mt genome of *O. denselamellosa* and molecular phylogeny will be useful in substantiating the molecular phylogeny for further evolutionary studies in relation to the conservation of Olympia oysters.

The histological features of the development process of oyster gonads are shown in Figure 4, with the SL ranging from 70.1 to 108.2 mm and the TW ranging from 85.4 to 240.4 g during the 10 months. In the resting stage, the undifferentiated stage of the gonads

is characterized by the absence of germ cells (Figure 4A). In the early gametogenesis stage, the oogonium and early vitellogenic oocytes or spermatogonia can be observed in follicles in the gonad (Figure 4B). In the advanced gametogenesis stage, the gonad follicles are filled with vitellogenic oocytes in a region at the edge of the follicles and spermatogonia and filled with spermatocytes in the center of the follicles (Figure 4C). In the mature gonad stage of males, females, and hermaphrodites, the cluster of spermatocytes and spermatozoa can be observed in the gonads of males, the follicles of gonads are filled with mature oocytes characterized by a discrete nucleus in females, and mixed mature gonads of males and females are observed in hermaphrodites (Figure 4D–F). In the spawned gonad stage, the gonad follicles of the spawn indicated that ovulation was released and oocytes were resorbed, smaller and relict spermatozoa were found, and the follicles were ruptured (Figure 4G). In hermaphrodites, the production of spermatozeugmata in multiple batches will enhance the success rate of fertilization when a batch of spermatozeugmata fails to reach females [18]. In this study, histological analyses indicated a discrepancy in gamete development in *O. denselamellosa* with a synchronous maturation of oocytes and an asynchronous development of spermatozoa in the gonads (Figure 4D,F). Previous studies have suggested that the production of spermatozeugmata by asynchronous gamete development will enhance fertilization success [18]. In *O. denselamellosa*, the discrepancy in gamete development may represent one of the strategies for successful fertilization before a brooding duration with fertilized eggs inside the body cavity and release of the larvae [11,19].

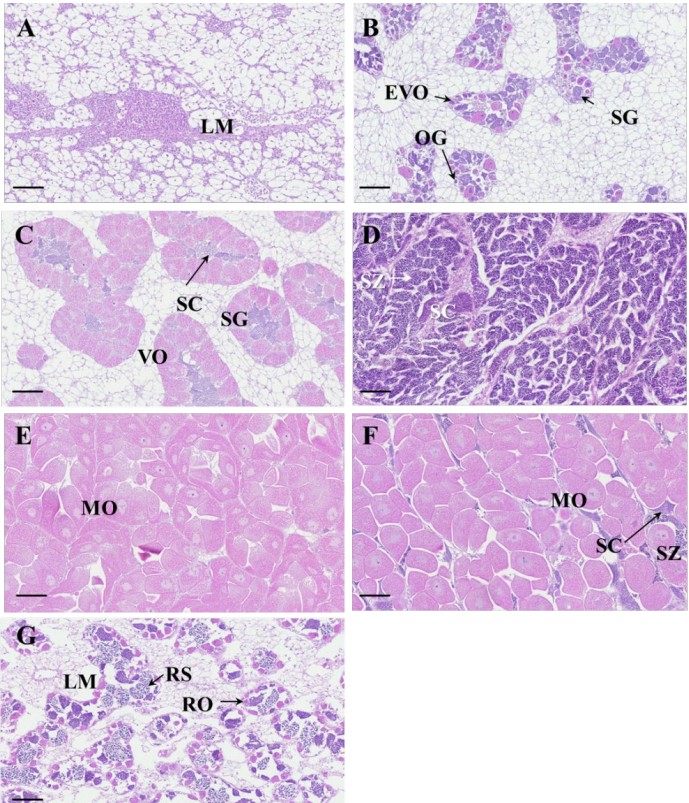

**Figure 4.** Photomicrographs of histological sections of *Ostrea denselamellosa* during gametogenesis. (**A**) Resting stage. (**B**) Early gametogenesis stage. (**C**) Advanced gametogenesis stage. (**D–F**) Mature gonad stage of males, females, and hermaphrodites. (**G**) Spawned gonad stage. EVO, early vitellogenic oocyte; LM, lumen; MO mature oocyte; OG, oogonium; RO, relict oocyte; RS, relict spermatozoa; SZ spermatozoa; VO vitellogenic oocytes.

In this study, the spawned gonads of *O. denselamellosa* were found from May onwards, dominating the first stage and mature gametogenesis when the water temperature reached

16.7 °C. In October, almost all individuals spawned gonads when the water temperature dropped below 23.2 °C. The frequency distribution of the gonad stage indicated that the spawning activity occurred from May to September (Figure 5). Based on the observations of the spawned gonad stage and resting stage in October, it was estimated that the stage of spawning activity from November to December was included in the resting period since previous studies suggested that the resting periods of gonad mature stages in *O. denselamellosa* occur from October to March in Goheung, along the southern coastline of Korea [32]. Water temperature is one of the key environmental factors that control the rate of gametogenesis in oysters [33,34]. In this study, the annual gametogenesis of *O. denselamellosa* can be described as the development of gametes in spring when the water temperature rises, maturation and spawning in summer, and resting in fall and winter when the water temperature decreases in Tongyeong, Korea (Figure 6). The duration of spawning activity of *O. denselamellosa* is similar to that of the Pacific oyster *Crassostrea gigas* in the coastal bays (Jaran Bay and Hansan-Koje Bay) of Korea from May to September [33,35]. The time of initiation of the spawning activity of *O. denselamellosa* is two months earlier than that of *Saccostrea kegaki* on Jeju Island from July to October [36]. The changes in gametogenesis of *O. denselamellosa* are similar to those in *C. gigas* in which the spawning activity occurs from June to September, with the spent condition and no gametes forming from October to February in Gosung Bay in Korea [34].

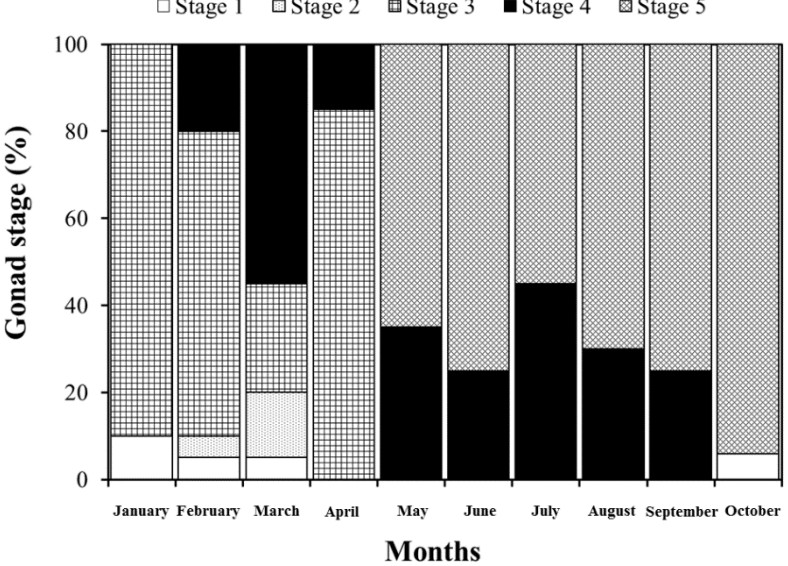

**Figure 5.** Monthly variation in gonad developmental stages of *Ostrea denselamellosa* collected from Tongyeong, South Korea, from January 2021 to October 2021.

The sex ratio during the study period is shown in Figure 6. The sex ratio of *O. denselamellosa* with an SL ranging from 70.1 to 108.2 mm was skewed toward females, which were most dominant accounting for approximately 75–80% of the total population from January to March, whereas hermaphrodites accounted for approximately 75–95% of the total population from April to October. Generally, broadcast oysters first mature as males and later change to females as they age [37]. The spermcasting oyster *O. edulis* has a highly skewed male-to-female ratio of 6:1 at an SL of approximately 50–70 mm [38,39]. The age of *O. denselamellosa* was estimated at approximately 2 years based on previous studies that reported that oysters reached approximately 80 mm in SL at the age of 2 years [40]. Therefore, the higher proportions of females and hermaphrodites than males in *O. denselamellosa* populations may be due to the development of relatively large oysters. The sex change patterns according to the lifespan during the present study were not clarified. Moreover, based on the phylogeny derived in this study, it would be beneficial to analyze the temperature-specific sex change patterns among different strains. Taken together, the reproduction

characteristics of *O. denselamellosa* indicated asynchronous gamete development patterns and the highest proportion of hermaphrodites. The spawning season was estimated to be from late spring to summer. However, the patterns of sex changes associated with the reproductive patterns were not determined. Thus, further studies in relation to the sex ratio and immature groups, as well as the mature and spawning groups associated with the lifespan of oysters, are required for elucidating the reproductive strategy.

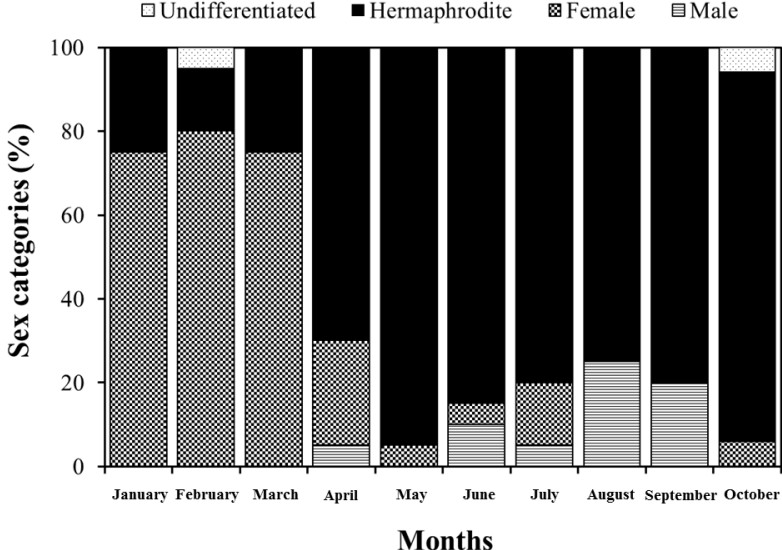

**Figure 6.** Distribution of sex categories of *Ostrea denselamellosa* collected from Tongyeong, South Korea, from January 2021 to October 2021.

In summary, this study determined the complete mt genome of the flat *oyster O. denselamellosa* collected from South Korea and performed phylogenetic analyses within Pteriomorphia. In addition, we confirmed the monthly changes in the gametogenesis, reproductive cycle, and sex ratio of *O. denselamellosa* from January to October 2021 in females. Overall, these findings will provide important information for further studies to help with the management and use of oyster resources.

**Author Contributions:** Data curation, Formal analysis, Writing—original draft, J.H.; Investigation, H.-J.K.; Conceptualization, S.-Y.O.; Conceptualization, Formal analysis, Writing—original draft, Y.-U.C. All authors have read and agreed to the published version of the manuscript.

**Funding:** This work was supported by the Korea Institute of Energy Technology Evaluation and Planning (KETEP) Ministry of Trade, Industry & Energy (MOTIE) of the Republic of Korea (No. 20203040020130).

**Institutional Review Board Statement:** All experiments were conducted in compliance with the guidelines of the Institutional Animal Care and Experimental Committee of the Korea Institute of Ocean Science and Technology (KIOST)", which approved the experimental protocol (No. 2021-03).

**Informed Consent Statement:** Not applicable.

**Data Availability Statement:** All data generated or analyzed during this study are available via the data repository of the KIOST. Requests for material should be made to the corresponding author.

**Acknowledgments:** The authors would like to thank to support of the Korea Institute of Energy Technology Evaluation and Planning (KETEP) and the Ministry of Trade, Industry & Energy (MOTIE) of the Republic of Korea (No. 20203040020130). Finally, we thank the editor and the anonymous reviewers whose comments greatly improved the manuscript.

**Conflicts of Interest:** The authors declare no conflict of interest.

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
