# Peer review of "Reproductive Characteristics of the Flat Oyster Ostrea denselamellosa (Bivalvia, Ostreidae) Found on the Southern Coast of South Korea"

_jmse, doi:10.3390/jmse10091326_

Round 1
Reviewer 1 Report
The authors of this paper sequenced the complete mitochondrial genome of the S. Korean strain of Ostrea danselamellosa before undertaking monthly field sampling of these flat oysters to determine their reproductive state over the course of 10 months. They found that the mt genome of the S. Korean strain of the oyster was very similar to that of the Chinese strain of the oyster and that spawning occurred when temperatures were higher during the summer and there were changes in the sex ratio throughout the sampling time. The study is clear and understandable and overall, the paper is well written and the English is of a good standard. The paper does however have some caveats which require clarification.
Using terms such as ‘seasonal dependant sex bias’ is unsuitable, as oysters were not sampled over the 4 complete seasons. Furthermore, as the reproductive cycle of this species has not been studied before in S. Korea, it would have been far better to sample over a much longer period- covering ideally 2 reproductive cycles. Were the environmental conditions encountered during in this study ‘typical’ of any given year? Additionally, the stress of being fished and taken to market immediately prior to this study will have undoubtedly stressed the animals- perhaps affecting their reproductive development long term. Did the authors take ‘time zero’ histological samples to determine the reproductive state of the animals prior to the start of the January 2021 sampling? Had they spawned due to the stress? Had they resorbed gametes due to the stress? Sampling oysters in situ every month may have been a better option.
The abstract requires some re-ordering. Start with something like “We investigated the seasonal reproductive pattern of the commercially and ecologically important species, Ostrea denelamellosa, to inform stock management strategies in South Korea. Firstly the complete mitochondrial…….”
“ Then, we investigated monthly changes in gametogenesis, reproductive cycle and sex ratio of oysters maintained in culture conditions (?) from January to October 2021, using histology. Histological analysis…..”
The similarity of the mt genome to the Chinese oyster strain is mentioned but not further discussed in the abstract (or the paper). Can the authors comment on the reproductive patterns seen in the Korean strain and how they compare to the Chinese strain?
Author Response
We greatly appreciate the valuable comments of reviewer on our work and thank them for their valuable suggestions on the manuscript. We have revised the manuscript as suggested by the reviewer. The corrected one was marked as the red in the revised manuscript. Comments and Suggestions for Authors The authors of this paper sequenced the complete mitochondrial genome of the S. Korean strain of Ostrea danselamellosa before undertaking monthly field sampling of these flat oysters to determine their reproductive state over the course of 10 months. They found that the mt genome of the S. Korean strain of the oyster was very similar to that of the Chinese strain of the oyster and that spawning occurred when temperatures were higher during the summer and there were changes in the sex ratio throughout the sampling time. The study is clear and understandable and overall, the paper is well written and the English is of a good standard. The paper does however have some caveats which require clarification. Using terms such as ‘seasonal dependent sex bias’ is unsuitable, as oysters were not sampled over the 4 complete seasons. Furthermore, as the reproductive cycle of this species has not been studied before in S. Korea, it would have been far better to sample over a much longer period- covering ideally 2 reproductive cycles. Were the environmental conditions encountered during in this study ‘typical’ of any given year? Additionally, the stress of being fished and taken to market immediately prior to this study will have undoubtedly stressed the animals- perhaps affecting their reproductive development long term. Did the authors take ‘time zero’ histological samples to determine the reproductive state of the animals prior to the start of the January 2021 sampling? Had they spawned due to the stress? Had they resorbed gametes due to the stress? Sampling oysters in situ every month may have been a better option. -> We absolutely agree with your comments. First, we have removed the word “seasonal” according to the reviewer’s suggestion in the revised manuscript. Also, as suggested by the reviewer, we will try to do next experiment to analyze the seasonal reproductive patterns in sampling oysters in situ every month. Considering the zero point control experiment, we also plan to include in the future studies. However, this study is a first step towards understanding how seasonal variation effects on reproductive patterns in native O. denselamellosa identified by mtDNA in South Korea. Therefore, we expect that reviewer try to focus on the novelty of experimental design and the adequate result (The first study to report the complete mt genome sequence and examine the seasonal reproductive pattern in native O. denselamellosa in South Korea) of this study. Thank you. The abstract requires some re-ordering. Start with something like “We investigated the seasonal reproductive pattern of the commercially and ecologically important species, Ostrea denelamellosa, to inform stock management strategies in South Korea. Firstly the complete mitochondrial…….” “Then, we investigated monthly changes in gametogenesis, reproductive cycle and sex ratio of oysters maintained in culture conditions (?) from January to October 2021, using histology. Histological analysis…..” -> Thank you for the comments. We have modified according to the reviewer’s suggestion in the revised manuscript. The similarity of the mt genome to the Chinese oyster strain is mentioned but not further discussed in the abstract (or the paper). Can the authors comment on the reproductive patterns seen in the Korean strain and how they compare to the Chinese strain? -> We agree to the reviewer’s comment. However, unfortunately, we can’t find the information of the reproductive patterns of O. denelamellosa Chinese strain. .

Reviewer 2 Report
Review
Paper title: Reproductive characteristics of the flat oyster Ostrea denselamellosa (Bivalvia, Ostreidae) found in the southern coast of South Korea
The authors applied next-generation sequencing technology to provide the complete mitochondrial genome of the flat oyster, Ostrea denselamellosa and studied seasonal changes in reproductive parameters of this endangered oyster species using histological analysis. They found a discrepancy in the gamete development with synchronous maturation of oocytes and asynchronous development of spermatozoa in gonads. They detected a seasonal bias in sex ratio. These data are an important contribution to the knowledge of the biology of O. denselamellosa in South Korea and may be used for further research and conservation measures.
All these reasons explain the relevance of the paper by Jeonghoon Han and co-authors submitted to "JMSE".
General scores.
The data presented by the authors are original and significant. The study is correctly designed and the authors used appropriate sampling methods. In general, statistical analyses are performed with good technical standards. The authors conducted careful work that may attract the attention of a wide range of specialists focused on benthic ecology and diversity.
Recommendations.
L 10-17. The authors should delete this section. It is not required for this journal.
The authors should analyze the contribution of environmental factors in driving reproductive parameters of Ostrea denselamellosa using correlation analysis.
Specific remarks.
L 3. Consider replacing “in the southern coast” with “on the southern coast”
L 18. Consider replacing “seasonal reproductive pattern” with “the seasonal reproductive pattern”
L 18. Consider replacing “O. denselamellosa” with “Ostrea denselamellosa”
L 29. Consider replacing “temperature” with “a temperature”
L 31. Consider replacing “sex-bias” with “sex ratio bias”
L 32. Consider replacing “pattern of” with “pattern in”
L 74. Consider replacing “reproduction cycle” with “the reproduction cycle”
L 75. Consider replacing “reproductive biology” with “the reproductive biology”
L 109. Consider replacing “from” with “from the”
L 123. Consider replacing “was collected” with “were collected”
L 131. Consider replacing “modification” with “modifications”
L 133. Consider replacing “category” with “categories”
L 211. Consider replacing “dominant” with “the dominating”
L 230. Consider replacing “spent condition” with “the spent condition”
L 247. Consider replacing “phylogeny” with “the phylogeny”
L 248. Consider replacing “temperature specific” with “temperature-specific”
L 264. Consider replacing “finding” with “findings”
L 309. “Ostrea denselamellosa” should be italicized.
L 321. “Ostrea angasi” should be italicized.
L 324. “Ostrea circumpicta” should be italicized.
Author Response
We greatly appreciate the valuable comments of reviewer on our work and thank them for their valuable suggestions on the manuscript. We have revised the manuscript as suggested by the reviewer. The corrected one was marked as the red in the revised manuscript. Comments and Suggestions for Authors Review Paper title: Reproductive characteristics of the flat oyster Ostrea denselamellosa (Bivalvia, Ostreidae) found in the southern coast of South Korea. The authors applied next-generation sequencing technology to provide the complete mitochondrial genome of the flat oyster, Ostrea denselamellosa and studied seasonal changes in reproductive parameters of this endangered oyster species using histological analysis. They found a discrepancy in the gamete development with synchronous maturation of oocytes and asynchronous development of spermatozoa in gonads. They detected a seasonal bias in sex ratio. These data are an important contribution to the knowledge of the biology of O. denselamellosa in South Korea and may be used for further research and conservation measures. All these reasons explain the relevance of the paper by Jeonghoon Han and co-authors submitted to "JMSE". General scores. The data presented by the authors are original and significant. The study is correctly designed and the authors used appropriate sampling methods. In general, statistical analyses are performed with good technical standards. The authors conducted careful work that may attract the attention of a wide range of specialists focused on benthic ecology and diversity. Recommendations. Line 10-17. The authors should delete this section. It is not required for this journal. -> Thank you for your comment. We have removed this section. The authors should analyze the contribution of environmental factors in driving reproductive parameters of Ostrea denselamellosa using correlation analysis. -> We absolutely agree with your recommendation. Currently, using this thesis as a starting point, we are in the process of constructing data to analyze the correlation between reproductive parameters of Ostrea denselamellosa and environmental factors (e.g., temperature and salinity). It is hoped that in the near future, like your suggestion, we will be able to propose a manuscript with more abundant data and more detailed information. Thanks again for your great suggestion. Specific remarks. L 3. Consider replacing “in the southern coast” with “on the southern coast” -> Corrected. L18. Consider replacing “seasonal reproductive pattern” with “the seasonal reproductive pattern” -> Corrected. L 18. Consider replacing “O. denselamellosa” with “Ostrea denselamellosa” -> Corrected. L 29. Consider replacing “temperature” with “a temperature” -> Corrected. L 31. Consider replacing “sex-bias” with “sex ratio bias” -> Corrected. L 32. Consider replacing “pattern of” with “pattern in” -> Corrected. L 74. Consider replacing “reproduction cycle” with “the reproduction cycle” -> Corrected. L 75. Consider replacing “reproductive biology” with “the reproductive biology” -> Corrected. L 109. Consider replacing “from” with “from the” -> Corrected. L 123. Consider replacing “was collected” with “were collected” -> Corrected. L 131. Consider replacing “modification” with “modifications” -> Corrected. L 133. Consider replacing “category” with “categories” -> Corrected. L 211. Consider replacing “dominant” with “the dominating” -> Corrected. L 230. Consider replacing “spent condition” with “the spent condition” -> Corrected. L 247. Consider replacing “phylogeny” with “the phylogeny” -> Corrected. L 248. Consider replacing “temperature specific” with “temperature-specific” -> Corrected. L 264. Consider replacing “finding” with “findings” -> Corrected. L 309. “Ostrea denselamellosa” should be italicized. -> Corrected. L 321. “Ostrea angasi” should be italicized. -> Corrected. L 324. “Ostrea circumpicta” should be italicized. -> Corrected.
